# Coupling Activity-Based Modeling and Life Cycle Assessment—A Proof-of-Concept Study on Cross-Border Commuting in Luxembourg

**Paul Baustert [1,2,\*], Tomás Navarrete Gutiérrez [2] , Thomas Gibon [2] , Laurent Chion [2],
Tai-Yu Ma [3], Gabriel Leite Mariante [4], Sylvain Klein [3], Philippe Gerber [3] and Enrico Benetto [2]**

1   Department of the Built Environment, Eindhoven University of Technology, 5612 AZ Eindhoven,
    The Netherlands
2   Luxembourg Institute of Science and Technology (LIST), 5, avenue des Hauts-Fourneaux,
    L-4362 Esch-sur-Alzette, Luxembourg
3   Luxembourg Institute of Socio-Economic Research (LISER), 11 Porte des Sciences,
    L-4366 Esch-sur-Alzette, Luxembourg
4   London School of Economics and Political Science (LSE), Houghton St, Holborn, London WC2A 2AE, UK
\*   Correspondence: paul.baustert@list.lu

**Abstract:** According to the Intergovernmental Panel on Climate Change (IPCC), in 2010 the transport sector was responsible for 23% of the total energy-related $CO_2$ emissions (6.7 $GtCO_2$) worldwide. Policy makers in Luxembourg are well-aware of the challenges and are setting ambitious objectives at country level for the mid and long term. However, a framework to assess environmental impacts from a life cycle perspective on the scale of transport policy scenarios, rather than individual vehicles, is lacking. We present a novel framework linking activity-based modeling with life cycle assessment (LCA) and a proof-of-concept case study for the French cross-border commuters working in Luxembourg. Our framework allows for the evaluation of specific policies formulated on the trip level as well as aggregated evaluation of environmental impacts from a life cycle perspective. The results of our proof-of-concept-based case study suggest that only a combination of: (1) policy measures improving the speed and coverage of the public transport system; (2) policy measures fostering electric mobility; and (3) external factors such as de-carbonizing the electricity mix will allow to counteract the expected increase in impacts due to the increase of mobility needs of the growing commuting population in the long term.

**Keywords:** life cycle assessment; activity-based modeling; policy analysis; sustainable mobility

## 1. Introduction

The environmental impacts of the transport sector are manifold. According to the Intergovernmental Panel on Climate Change (IPCC) in 2010, it was responsible for approximately 23% of total energy-related $CO_2$ emissions (6.7 $GtCO_2$) worldwide [1]. The World Health Organization (WHO) holds the transport sector responsible for a large proportion of urban air pollution, attributing an estimated 3.7 million premature deaths to ambient (outdoor) air pollution [2]. In addition, internal combustion engine vehicles contribute to fossil fuel depletion, while commercially available electric vehicles require the mining of lithium, cobalt and rare earth metals related to both social and environmental issues [3].

In its summary for policy makers [4], the IPCC has identified several potential measures to reduce transport-related $CO_2$ emissions including infrastructure development and behavioral change. They argue that a reduction of 15–40% could be achieved in 2050 [1].

The government of the Grand-Duchy of Luxembourg seems well aware of the challenges surrounding transport and has made transforming the national transport sector one of their priorities. Several strategic documents have been published over the course of the last decade. This includes technical and economic studies, e.g., investigating the challenges of implementing a national charging infrastructure to foster electric mobility [5]. Two studies specifically focus on the sustainable transition of the entire transport system. The first study [6] formulates four specific objectives for 2020: (1) better articulation between territorial development and mobility; (2) 25% of daily trips through soft mobility; (3) 25% of motorized trips by public transport; (4) promote alternative use of the car. Following this initial study, [7] evaluates if the former objectives are likely to be achieved, drawing mixed conclusions, while formulating a new set of objectives for 2025 related to: (1) the modal shares of home-work commuting (e.g., increase by 50% the number of public transport, or increase soft mobility for commuting under 5 km); (2) increasing the occupation rate of private vehicles; (3) the modal shares of home-school trips; (4) the attractiveness of public transport (e.g., reducing delays and cancellations). Overall, one can conclude that the new objectives for 2025 are much more compatible with commuter daily activity patterns, distinguishing between different specific challenges (e.g., home-work commuting during peak hours).

However, these studies and objectives are not an end in itself but clearly linked to sustainability goals as illustrated in [7]: (1) the de-carbonization of the transport sector to achieve the ambitious national targets fixed in the Paris climate agreement namely a 40% reduction of greenhouse gas (GHG) emissions; (2) improving air quality according to EU directives with a pledge of 83% reduction of nitrogen oxide (NOx) emissions and of 40% reduction in fine particulate matter (PM2.5) emissions; (3) contributing to the third industrial revolution formulated in another strategic document [8]; (4) "transport for everyone" towards a more inclusive transport system.

While these goals should be commended, the effectiveness of different policy measures designed to achieve them must be tested, since their impact might not be self-evident, potentially leading to rebound effects and often implying trade-offs [9,10]. This is especially true for policy measures aiming at mitigating climate change, where some options could result in a shifting of the burden rather than actual mitigation. Additionally, it can be misleading to consider only local or use phase impacts, without considering the whole chain of life cycle impacts that go along with transport systems. Therefore, we focus the present work on life cycle assessment (LCA) based approaches, allowing to consider all product or service life phases from cradle-to-grave [11].

An extensive systematic literature review was conducted using Scopus and Google Scholar in June and July 2019. The keywords "(Life Cycle Assessment) AND (transportation) AND (policies)" were used on Scopus. The initial 256 identified records were combined with 11 records found from a previous Google scholar search. All articles were screened, first using title and abstract then by full text review. The Preferred Reporting Items for Systematic Reviews and Meta-Analyses (PRISMA) flowchart [12] was used to document the process and all articles and exclusion criteria can be found in the supplementary materials. Finally, 28 articles were retained and categorized according to the investigated policy measure scenarios, distinguishing vehicle fleet composition scenarios, mode choice scenarios, operation or management scenarios and infrastructure scenarios. Figure 1 shows a Venn diagram of the reviewed literature classified by the policy measure scenarios.

Evaluating policy measure scenarios aiming at evaluating a fleet from a life cycle perspective is the most addressed topic in the reviewed literature. The implementation of policy measure ranges from various types of scenarios (e.g., fleet shares, comparing powertrains in specific settings, etc.) [13–29], over more complex fleet modeling (e.g., using cohort models) [30,31], to fine grained agent based modeling using decision trees [32]. While scenarios and stock models allow to evaluate policy targets (e.g., future target fleet or market shares), the agent-based modeling approach allows to evaluate specific policy measures such as subsidies and their impact on consumer choices.

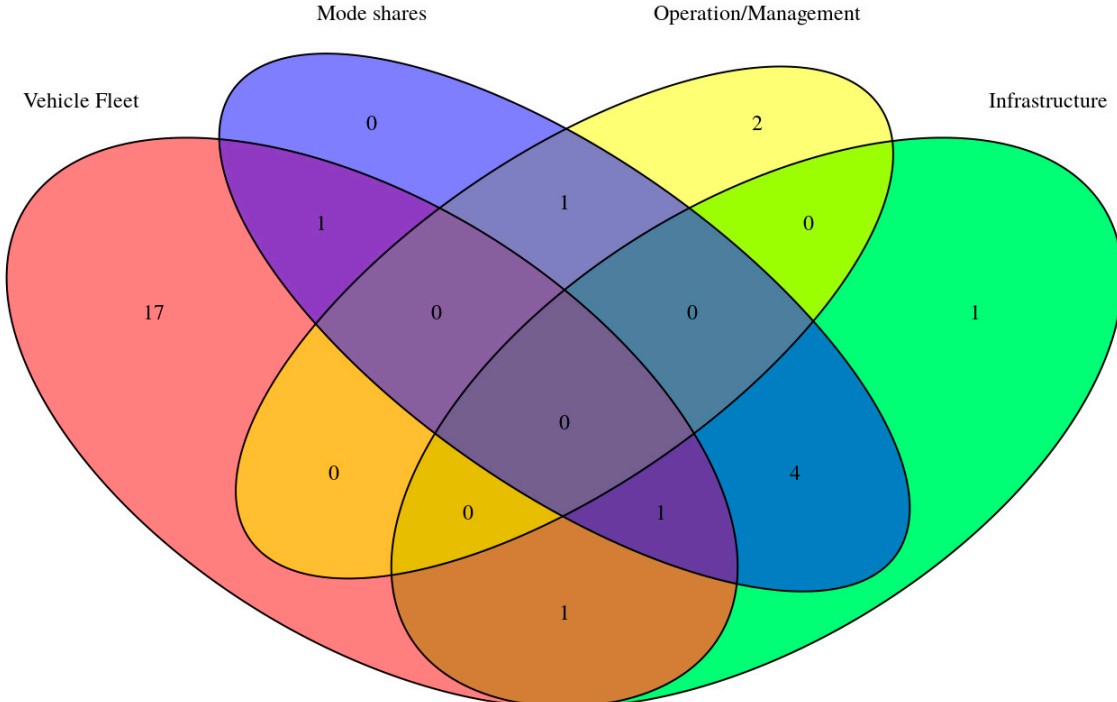

**Figure 1.** Venn diagram of reviewed articles classified according to policy measure scenarios. Numbers in the diagram are the quantity of articles found belonging to a set or overlap.

Another policy-related aspect taken into account in several studies are the mode shares (e.g., between personal car usage and public transport). In all of the reviewed studies that investigated mode shares, some kind of scenario analysis was conducted [14,15,33–37]. In some cases, hypothetical values were compared [37], while in other cases, more detailed analyses were conducted and sensitivities to assumptions evaluated [14]. Potentially, mode choice behavior could be investigated at the individual level using microsimulations; however, no such LCA study was found.

Only a few studies evaluate operation and management measures from a life cycle perspective. Studies investigate measures such as parking management (e.g., pricing) [37], public transport operation strategies (e.g., real time control tactics) [38] and traffic incident management [39]. These studies use simulations [39] real-world data [38] or scenarios and surveys [37] depending on the investigated operation and management strategy.

Finally, policy measures regarding infrastructure are evaluated by several studies often in combination with mode choice. This is not surprising as often the purpose of large-scale infrastructure projects is to increase the usage of some mode categories (e.g., public transport), while reducing the mode share of others (e.g., private vehicles). This concerns mainly planned or existing high-speed or light-rail line projects [14,33–36,40] and in one case infrastructures to foster electric mobility [28].

Our review of literature reveals that while there are several studies investigating various policy measure scenarios from a life cycle perspective, they tend to be focused on the vehicle fleet, while only a few investigate other aspects. While mode share policy targets are investigated, this is mainly done using scenarios without taking into account the impact of concrete measures on individual mode choice behavior.

However, our review of relevant strategic policy reports for Luxembourg has shown that mobility policy evaluation requires approaches that allow for fine granularity with regard to travel behavior, as goals are formulated on the trip level (e.g., the modal shares of home-work commuting). An interesting approach to evaluate the sustainability of transport policies discussed in the literature, e.g., [41], is activity-based travel demand modeling (ABM). ABMs allow to generate individual activity patterns, taking into account constraints in time and space as well as relations between activities and travels for each individual [42]. Usually, ABMs are composed of several model components, such as a

population synthesis model to generate a synthetic population for the study region, a set of choice models to generate the conducted activities, the activity durations, the activity locations and travel modes, and a network supply model generating the performance of the transportation network [42]. ABMs promise some distinct advantages compared to conventional four-step models [43]. Coupling this type of modeling with LCA would allow evaluating targeted policy action based on individual activity patterns and assess the related environmental impacts.

Therefore, this paper aims at: (1) demonstrating the feasibility to couple such an ABM with the general LCA framework, which is, to the best of our knowledge, a novel and unique modeling framework; (2) illustrating how this framework can be used to assess the success of policy action in reaching specific targets on the trip level; (3) illustrating how to assess the life cycle impacts of the entire transport systems for different policy scenarios.

In Luxembourg, the large number of cross-border commuters coming from all neighboring countries (overall 180,000 from France, Belgium and Germany), making up around half of the national workforce [44], are a noticeable challenge for the regional transport system. This study focuses on cross-border commuters in general and the largest part of cross-border commuters coming from France in particular, building on prior work done for this subset of the commuting population. To evaluate specific policy actions, options to increase the modal share of public transport for this population are investigated. In addition, the impact of fostering electric mobility on the system's environmental impact will be evaluated. The choice of these two policy related scenario aspects is motivated by the fact that: (1) both are explicitly mentioned as important bricks of a future sustainable transport system in the consulted documents and; (2) that they reflect both a technical solution electric vehicles (EVs) and a behavioral change (mode choice toward public transport) induced by infrastructure and operation measures.

The remainder of this article will adopt the following structure: Section 2 will introduce the adopted methodologies to generate the individual activity patterns and life cycle impacts; Section 3 will introduce the chosen case study; Section 4 will present the results which will be discussed in Section 5.

## 2. Materials and Methods

In this section, the integration of ABM with LCA in the context of mobility policy support is introduced. First, the relevant parts of the ABM used to derive activity patterns for a population, focusing on the individual mode choice, are presented. Furthermore, the transformation of these activity patterns into the system's demand, using simple aggregation and a cohort model to derive the shares of different powertrains in the car fleet for each year, are introduced. Finally, the coupling of both modeling approaches, where the system's final demand is at the interface of the ABM and LCA part of model, is presented.

### 2.1. Activity-Based Modeling of Travel Demand

To generate the individual activity patterns for a given population, an ABM was developed in [45] for the French cross-border commuters working in Luxembourg, building on a survey conducted in the study region. The model is composed of interdependent econometric discrete choice models to yield the activity types, activity and travel durations, activity locations and mode choices, thus a full activity pattern for each individual of the population. Figure 2 is an example of an activity pattern, as generated by the developed ABM.

While the models used to derive the activity types, durations and locations are identical to those described in [45], the models generating the mode choices were adapted for the purpose of updating the activity patterns for different policy scenarios. Specifically, the mode choice models used for the present work are based on a multi-nominal logistic (MNL) modeling approach:

$$log\left(\frac{P(M_{ij} = m)}{P(M_{ij} = ref)}\right) = \beta_{0m} + \sum_{k=1}^{K} \beta_{km}X_{ki} + \alpha_{1m}Y_{ij} + \alpha_{2m}DC_{ij} + \alpha_{3m}NT_{ij} + \sum_{t=1}^{T} \alpha_{tm}Z_{tod} + \epsilon_{mij} \quad (1)$$

where $M_{ij}$ is the mode choice for the trip to the $j_{th}$ activity undertaken by the $i_{th}$ individual, with the choice being one of seven possible alternatives (car, bus, train, multi-modal with car, multi-modal without car, soft mobility and other) represented by $m$. $X_k$ are $K$ survey response variables for the $i_{th}$ individual, $Y_{ij}$ is the predicted type of the $j_{th}$ activity undertaken by the $i_{th}$ individual, $NT_{ij}$ is the predicted number of trips undertaken by the $i_{th}$ individual over his entire activity pattern and $DC_{ij}$ is an ordinal variable indicating the departure time of the trip to the $j_{th}$ activity by the $i_{th}$ individual. $Z_{tod}$ are the travel times for mode $T$ travel modes of the specific origin destination (OD) pair. The alternative specific terms $\epsilon_{mij}$ are extreme-value distributed. The reference category (*ref*) is "other", where the $P(M_{ij} = m)$ is the choice probability of mode m and $P(M_{ij} = ref)$ is the choice probability of the reference category. If we exponentiate both the left and right hand side of Equation (1) we can solve for $P(M_{ij} = m)$ for each $m$.

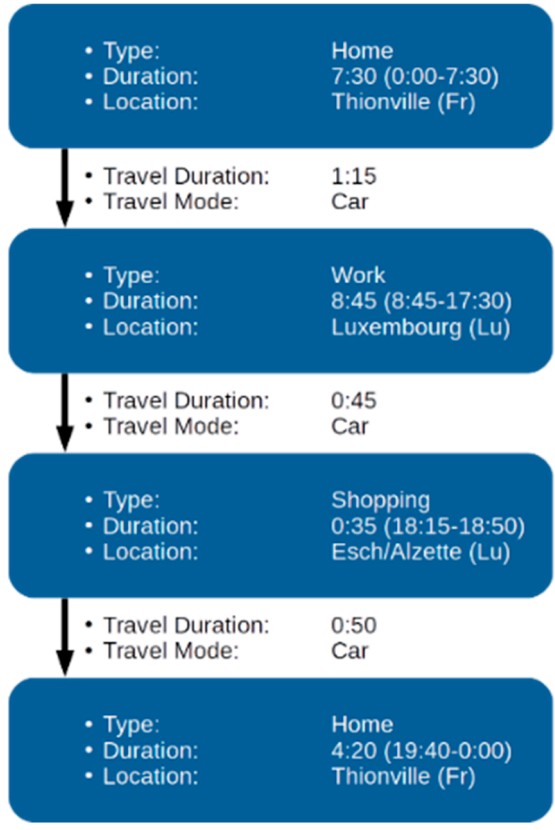

**Figure 2.** Daily activity pattern, adapted from [45].

As far as a population with the necessary response variables and the travel time data for each OD pair is provided, the ABM allows to generate one activity pattern for each individual in the population similar to Figure 2. Thus, the output of the model is the set of all activity patterns for the entire population.

### 2.2. Deriving the System's Final Demand Vector

For the purpose of this article, the emphasis is placed on the mode choice and travel modes, while other aspects of the activity pattern are assumed to be unaffected by policy decisions. We will refer to the aggregated results over all activity patterns for one simulation of the ABM as $r_M$, where $r$ is the result vector for a concrete instance $M$ of the ABM, defined by a specific scenario and year.

As only the aggregated travel distances for each transport mode are of interest for the LCA part, to arrive at $f(r_M)$, the system's demand for transport processes in person kilometers (pkm), needs to be derived. To this end, an origin-destination matrix (OD-matrix) containing travel distances for all

available transport modes between all possible OD pairs can be used. Finally, aggregating the travelled distances over the investigated population and individual activity patterns allows to arrive at the pkm for each modeled transport mode contained in $f(r_M)$ as shown in (2), for a concrete instance $M$ of the ABM:

$$f(r_M) = \left[ \overbrace{d_{EV} \; d_{PHEV} \; d_{DIE} \; d_{GAS}}^{car\ power\ trains} d_{BUS} \; d_{TRAIN} \cdots \right]^T \tag{2}$$

To estimate the share of different powertrainpowertrains and emission standards in the car fleet, a stock-flow cohort model was adopted, which can be considered a simplified version of the model presented in [46]. Given an initial age distribution of vehicles for each powertrain (and emissions standard) type $i$ over 30 age classes $j$, the stock of vehicles for each age class and powertrain type (and emission standard) for year $n + 1$ can be derived as follows:

$$A_{i,j+1}^{n+1} = A_{i,j}^{n}\left[1 + \alpha_{i,j+1}\right], \tag{3}$$

where $\alpha$ is derived based on the cumulative survival probability curves presented in figure five of [46] for the different types of powertrains and age classes. Each year, new vehicles are added to the first age class in order to satisfy the overall demand for vehicles by the population and are distributed among powertrains based on market share for the specific year. Using the resulting fleet shares of the different powertrains, each individual in the population is randomly attributed a car powertrain for a given year, regardless of his mode choice record. For one simulation of the ABM part of the model, one can thus derive $f(r_M)$, distinguishing various powertrains and emission profiles.

### 2.3. Integration with Life Cycle Assessment (LCA)

Baustert and Benetto [47] discussed the integration of agent-based modeling (of which activity-based modeling can be seen as a specific example) with LCA, distinguishing two different computational implementations available in literature.

The integration in this study will follow a computational structure where the final demand vector $f$ is at the interface of both models, which can be formulated according to the matrix algebra and general computational structure of LCA [48], where each element in the final demand vector $f$ represents the demand of the transport system for a life cycle inventory (LCI) process (i.e., passenger car transport) and some or all elements become a function of the result vector $r_M$ of the ABM, as shown in (2).

Ultimately the computational structure of [48] yields the environmental interventions vector $g$ containing, e.g., greenhouse gas emissions, particulate matter emissions, etc.:

$$g(r_M) = BA^{-1}f(r_M), \tag{4}$$

where $A$ is the technology matrix, containing the technical coefficients of the various processes involved in the transportation system, and $B$ is the intervention matrix, containing the environmental flows of each of these processes. Next, the environmental impact vector $h$ can be derived:

$$h(r_M) = Qg(r_M), \tag{5}$$

where $Q$ is the characterization matrix, made of fixed coefficients linking the midpoint impact categories with the environmental interventions. Figure 3 shows a schematic overview of the coupled model proposed in this article.

For each of the two scenarios presented in the following case study section a set of final demand vectors is derived:

$$F_{BAU} = [f(r_{BAU,2015}), \ldots, f(r_{BAU,2025})], \tag{6}$$

$$F_{GREEN} = [f(r_{GREEN,2015}), \ldots, f(r_{GREEN,2025})], \tag{7}$$

where we can derive the environmental impacts according to (4) and (5) for each element in $F_{BAU}$ and $F_{GREEN}$.

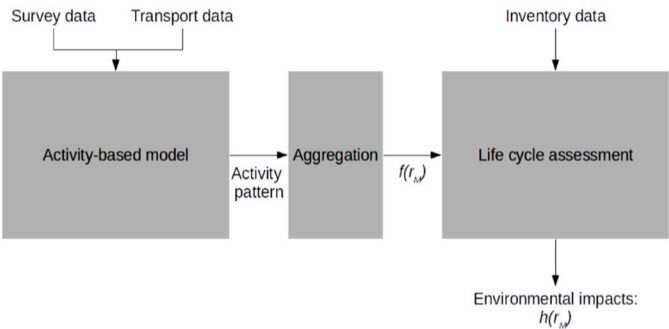

**Figure 3.** Schematic overview of the proposed coupled activity-based travel demand modeling (ABM)/ life cycle assessment (LCA) model.

## 2.4. Simulator Implementation

To implement the described mathematical model, the ABM part was coded in R. The "mlogit" package [49] was used to implement both the estimation of the parameters in (1) and prediction of mode choices done for the case study. The ABM is available in supplementary materials.

The LCA integration part creating the final demand vector (2) and performing the LCA related calculations (4) and (5) was coded in Python. The "brightway2" framework [50] was used for the LCA part, while the integration had to be developed from scratch. The life cycle inventories (LCI) used to conduct the LCA were created specifically for this study, adapting existing inventories from the database for LCA ecoinvent version 3.5 "cutoff". This database and the derived inventories are subject to intellectual property rights (IPR) protection and are not disclosed with this paper.

Finally, the cohort model is implemented using a simple spreadsheet applying Equation (3).

## 3. Case study

The case study investigates how different levels of investments by decision-makers can influence the activity pattern of the French cross-border commuters working in Luxembourg, as well as the potential impact of a raise of the share of EVs in the car fleet. The study period is defined from 2015 to 2025.

An initial synthetic population and their activity pattern is generated based on the survey conducted by the Luxembourg Institute of Socio-Economic Research (LISER) described in [51,52] and the ABM described in [45], where the mode choice remains undetermined at first.

### 3.1. Policy Scenarios

As mentioned in Section 1, policy makers are trying to reduce the environmental burden associated to commuter mobility through various measures. At the current stage, the model considers two main policy measures: (1) improving the quality of the public transport system by offering more adequate alternatives that improve coverage and reducing travel times; (2) increasing the share of low emission vehicles in the car fleet. In this section the implementation of these two policy measures is presented for the case study. Thus, we aimed at demonstrating how our framework can evaluate both technical and behavioral solutions.

### 3.1.1. Evolution of the Transport System

Before introducing the needed data and OD matrix generation for the scenarios, the underlying rationale for this part of the policy scenario needs to be made explicit. As illustrated in [6,7] policy makers are trying to increase the share of commuters using public transport, among other measures, by improving infrastructure and offer. This includes new trans-border bus lines, increasing the frequency

of trains by rail doubling or decreasing travel times by opening new peripheral train stations. Our model can take such measures into account as long as they can be related to some of the variables in (1), i.e., $Z_{tod}$.

To derive realistic scenarios regarding the evolution of the transport system, raw data are required reflecting the state of the system (i.e., travel times and distances) at the beginning of the study period in 2015. To this end, current General Transit Feed Specification (GTFS) data of the study region provided by the Luxembourgish open data platform [53] are used together with OpenStreetMap GIS data of the road and rail network in the study region acquired from the "Geofabrik" service [54], to mount an open-trip-planner (OTP) server. Using the centroids of the 916 communes in the study region, the OTP is used to derive travel times and transport distances for single modes (car, bus train) as well as multi-modal combinations with and without car. In total, 916 × 916 OD pairs are evaluated.

This original OD matrix is then used to generate yearly matrices for the length of the study period reflecting two distinct scenarios: (1) business-as-usual (BAU) reflecting "moderate" yearly changes regarding coverage and speed of the study region; (2) ambitious environmental measures (GREEN) reflecting "high" yearly changes regarding coverage and speed of the study region.

More specifically, to address the coverage of public transport, travel times and distances are derived for OD pairs for which the OTP did not find an adequate link. This is how we implemented the first part of policy measures to improve the quality of public transport (see Section 3.1). We refer to this implementation as the coverage algorithm. To address the travel times of public transport, existing connections are updated by reducing the observed travel time. We refer to this as the speed algorithm. To generate a set of OD matrices with coherence in time, for a given scenario these algorithms were recursively applied on the reference OD matrix over the course of the study period. Both algorithms are focused on origins in the French part of the study region for which the National Institute of Statistics and Economic Studies of the Grand Duchy of Luxembourg (STATEC) reports at least: (i) a certain threshold amount of cross-border commuters and (ii) a set of destinations in the Luxembourgish part of the study region reflecting major economic areas (i.e., the city of Luxembourg and Esch-sur-Alzette), as well as locations of co-working hubs targeting French cross-border commuters (i.e., Bettembourg and Foetz).

To derive the travel times and distances of newly added connections by the coverage algorithm, the average ratios of travel time and distance of all bus connections to the car data in the original matrix is used. To derive at the travel times of the updated connections by the speed algorithm, different quantiles of the ratios of travel times in the original matrix are used for both scenarios. The implementation of both algorithms for the identified origins is spread evenly over the study period, prioritizing origins with high amounts of cross-border commuters. While the coverage algorithm is applied only for flexible transport modes (i.e., bus and public transport with car), the speed algorithm is applied also for non-flexible modes (i.e., train). Table 1 gives an overview of the parameters used for each algorithm in each of the scenarios.

**Table 1.** Parameters of scenarios used to implement the evolution of the transport system. CBC standing for cross border commuter and NA standing for not available

| Scenario | Speed Algorithm | Coverage Algorithm |
|---|---|---|
| BAU | CBC threshold: 1000<br>Speed ratio quantile: 0.05 | CBC threshold: 450 |
| GREEN | CBC threshold: 450<br>Speed ratio quantile: 0.01 | CBC threshold: NA |

### 3.1.2. Market Penetration of EVs

Before introducing the scenario definition with regard to EVs, again, the underlying rationale needs to be made explicit. As illustrated in [7,8], policy makers have set ambitious goals to phase out internal combustion engine vehicles (ICEVs) by 2050. Specific policy measure such as new charging

infrastructures for EVs and subsidies are implemented to reach this target. While the impact of such measure on the individual commuter are not directly modeled, the potential impact of reaching this ambitious target can be assessed by contrasting two consistent storylines describing the evolution of market shares.

The two scenarios for electric vehicle (EV) market shares are: (1) the French Environment and Energy Management Agency (ADEME) one, which is inspired by a report of the French Agency for Environment and Energy Management [55]; (2) the TIR one which is inspired by the above mentioned s (TIR) strategy study for the Grand Duchy of Luxembourg [8]. Figure 4 shows the respective market shares over a period from 2015 until 2050 taking also the most recent sales statistics published by the National Institute of Statistics and Economic Studies (INSEE) into account. Four powertrains are distinguished: gasoline, diesel, battery electric vehicles (BEV) and plug-in hybrid electric vehicles (PHEV). Other powertrains are disregarded as no reliable data regarding market shares was available.

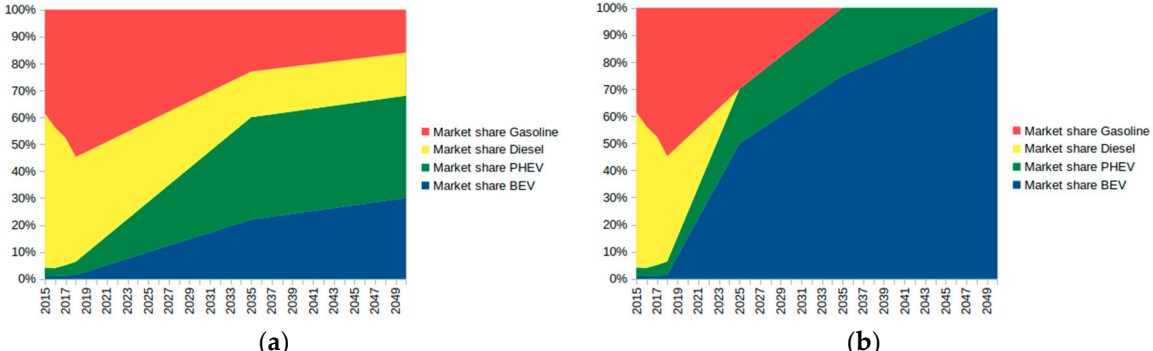

(a) (b)

**Figure 4.** Market shares reflecting different visions with regard to the EV market penetration where from 2015 until 2018 most recent sales statistics are considered. (**a**) ADEME-inspired market shares; (**b**) third industrial revolution (TIR)-inspired market shares.

Our cohort model allows to arrive at the car fleet compositions, where assumptions are made about the initial age distribution (uniform) and the total demand for French cross-border commuters is based on population projections and their average car ownership issued by regional authorities [56].

*3.2. External Factors*

In addition to the policy scenarios, exogenous factors influence the evolution of environmental impacts in the study region. Luxembourgish economy has one of the highest economic growth rates of the European Union, with equally high growth rates of the resident population and of cross-border commuters [57]. STATEC has investigated the evolution of cross-border commuters over the past decades. Figure 5 shows the steady increase of cross-border commuters from the neighboring countries, as well as current projections which are based on medium [58] and long term [44] macroeconomic and demographic modeling. In order to distinguish between French cross-border commuters and commuters from Belgium and Germany, we assumed that the current distribution between regions (where around 50% of all cross-border commuters come from France) remains stable until 2060. Especially, for long term projections, the related uncertainties can be considered to be high, therefore high and low projections were added based on the estimations of [44], starting 2023 (which is the last year of the medium term projection in [58]). The assumptions regarding long term projections were used to derive the number of cross-border commuters from 2024 onwards table number four [44] (p. 37).

These scenarios of population growth are considered in the model, where the number of French cross-border commuters increases by around 50% over the course of the study period (2015–2025). This raises the questions if individual or even combinations of the planned policy measures allow

to counteract the expected increase in impacts due to the increase of mobility needs of the growing commuting population.

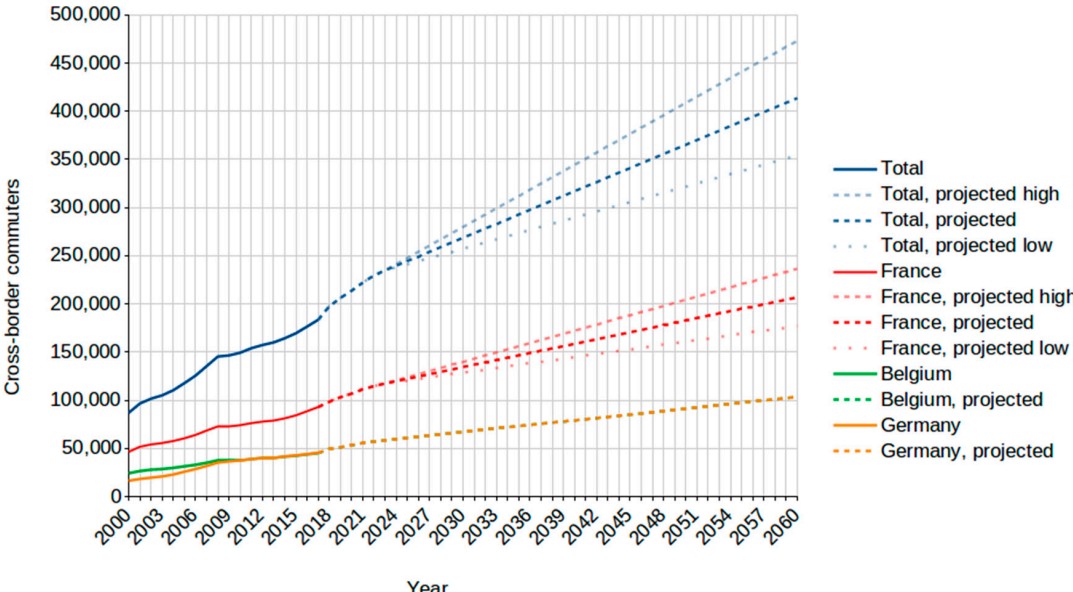

**Figure 5.** Historical data and projections of cross-border commuters from neighboring countries to Luxembourg.

A second (partially) external factor is the evolution of technology (i.e., vehicle consumption) and background processes, (i.e., electricity mixes), over the course of the study period. To this end, the life cycle inventory (LCI) processes (described in Section 3.3.2) are parameterized to change each year to reflect technological advancements. In order to assess the impact of this factor, all scenarios are run with the following two databases: (1) with a constant database where all process parameters remain fixed at the 2015 level; (2) an evolving database where process parameters evolve over time.

Parameters have been introduced for electricity mixes, and vehicle consumption. Other potential parameters were screened, e.g., vehicle light-weighting, but were deemed not significant enough regarding the short simulation window of 10 years. Electricity mixes have been modeled as described in Table 1, and are assumed to evolve linearly from 2015 to 2025. Initial values (2015) are extracted directly from ecoinvent 3.5 in its "cutoff" version (which, to simplify, adopts a cradle-to-gate approach), final values (2025) were assumed with the following rationale:

- Luxembourg: increase in natural gas and wind power generation domestically, decrease in imports (namely from the coal-intensive German mix),
- France: the objective of reaching 50% of nuclear power after 2030 in the national mix entails a share of about 64.5% in 2025 from this source, replaced by wind and hydro power.

*3.3. Life Cycle Assessment (LCA)*

3.3.1. Goal and Scope

The goal the life cycle assessment is to evaluate the environmental impacts of all mobility systems used to meet the demand of the cross-border commuters residing in France and working in Luxembourg for the period 2015–2025. These systems encompass: private vehicles and public transportation, namely trains, buses and tramways. Infrastructure (roads, railways …) are included in these systems. The functional unit for this study is to "meet the daily travel demand by French cross-border commuters for 2015–2025 in Luxembourg". The system modeled includes: the production of vehicles and rolling stock, the infrastructure they use (roads and rails), their maintenance, the fuels and energy carriers' supply chains, as well as the use and end-of-life phase.

### 3.3.2. Life Cycle Inventory (LCI)

The LCA part of the model is based on region specific datasets which considers the manufacturer data on trains and buses used by the region operators. For the sake of the development of the Proof-of-Concept, background LCI data are taken from the ecoinvent 3.5 "cutoff" database. The use of consequential datasets will be explored in a further development of the model. Public transport processes that use electricity are all connected to Luxembourg's high-voltage mix. Electric buses are assumed to adopt a mix of two charging approaches: opportunistic (charge may occur during operation, which allows for a lighter battery) and overnight (charge only occurs at night, which requires a larger battery). BEVs and PHEVs are charged in France, with a low-voltage mix.

Inventories are parameterized, mainly according to the assumptions that: electricity mixes of Luxembourg and France evolve between 2015 and 2025 (see Table 2), and that the average efficiency of the vehicle fleet increases. For the latter consideration, specific consumptions of ICE passenger vehicles change with time, as described in Table 3. These changes are modeled linearly, and inventories are therefore recalculated for every time step of the simulator, after which they are scaled up according to the specific distance demand, per mode, of that time step. The life cycle impact assessment (LCIA) is calculated in a next phase, as described in the following section.

**Table 2.** Electricity mixes for Luxembourg and France, as modeled, as well as their global warming potential.

| (%) | | Luxembourg | | France | |
|---|---|---|---|---|---|
| | | **2015** | **2025** | **2015** | **2025** |
| Electricity Source | Photovoltaics | 0.0 | 0.0 | 0.8 | 0.8 |
| | Coal | 0.0 | 0.0 | 1.6 | 0.0 |
| | Hydro | 12.2 | 12.0 | 12.9 | 19.8 |
| | Natural gas | 0.0 | 20.0 | 0.6 | 0.0 |
| | Oil | 0.0 | 0.0 | 0.1 | 0.0 |
| | Wind | 0.8 | 10.0 | 3.0 | 14.9 |
| | Nuclear | 0.0 | 0.0 | 78.5 | 64.5 |
| | CHP * | 13.9 | 15.0 | 1.1 | 0.0 |
| | Imports | 73.1 | 43.0 | 1.5 | 0.0 |
| (g $CO_2$ eq/kWh) | | | | | |
| GWP100 | High voltage | 584 | 362 | 42.9 | 14.2 |
| | Low voltage | 597 | 372 | 49.7 | 19.1 |

\* Combined heat and power.

**Table 3.** Life cycle inventory (LCI) processes and data sources. The basic source for each inventory is the ecoinvent 3.5 cutoff database.

| Mode | Inventory and Modifications | |
|---|---|---|
| | *from* | *to* |
| Train | *Transport, passenger train {RER}*<br>Assumptions:<br>• Train produced in Europe<br>• Uses Luxembourg's electricity | *Transport, passenger train {LU}* |
| Bus (diesel) | *Transport, regular bus {RoW}*<br>Assumptions:<br>• Lifetime: 1 million km<br>• Average occupancy of 14<br>• Average consumption of 38 l/100 km | *Transport, regular bus, diesel, articulated {LU}*<br>*Transport, regular bus, diesel, not articulated {LU}* |

**Table 3.** *Cont.*

| Mode | Inventory and Modifications | |
|---|---|---|
| | *from* | *to* |
| Bus (electric) | *Transport, regular bus {RoW}*<br><br>Assumptions:<br><br>• Lifetime: 1 million km<br>• Average occupancy of 14<br>• Average consumption of 1.3 kWh/km<br>• 60% opportunity charging (150 kWh battery)<br>• 40% overnight charging (250 kWh battery)<br>• All charged in Luxembourg<br><br>Modeled after Volvo 7900 [59] | *Transport, regular bus, electric, opportunity charging, not articulated {LU}*<br>*Transport, regular bus, electric, overnight charging, not articulated {LU}* |
| Car (diesel) | *Transport, passenger car, large size, diesel, EURO 3 {RER}*<br>*Transport, passenger car, medium size, diesel, EURO 3 {RER}*<br>*Transport, passenger car, large size, diesel, EURO 4 {RER}*<br>*Transport, passenger car, medium size, diesel, EURO 4 {RER}*<br>*Transport, passenger car, large size, diesel, EURO 5 {RER}*<br>*Transport, passenger car, medium size, diesel, EURO 5 {RER}*<br><br>Assumptions:<br><br>• Lifetime: 250,000 km<br>• Large (2000 kg, 7.0–5.0 L/100 km average over 2015–2025)<br>• Medium (1600 kg, 5.0–4.0 L/100 km average over 2015–2025)<br>• EURO 6 diesel vehicles are adapted from EURO 5 by changing NOx emissions<br><br>Small is not considered | *Transport, passenger car, large size, diesel, EURO 3 {RER}*<br>*Transport, passenger car, medium size, diesel, EURO 3 {RER}*<br>*Transport, passenger car, large size, diesel, EURO 4 {RER}*<br>*Transport, passenger car, medium size, diesel, EURO 4 {RER}*<br>*Transport, passenger car, large size, diesel, EURO 5 {RER}*<br>*Transport, passenger car, medium size, diesel, EURO 5 {RER}*<br>*Transport, passenger car, large size, diesel, EURO 6 {RER}*<br>*Transport, passenger car, medium size, diesel, EURO 6 {RER}* |
| Car (gasoline) | *Transport, passenger car, large size, petrol, EURO 3 {RER}*<br>*Transport, passenger car, medium size, petrol, EURO 3 {RER}*<br>*Transport, passenger car, large size, petrol, EURO 4 {RER}*<br>*Transport, passenger car, medium size, petrol, EURO 4 {RER}*<br>*Transport, passenger car, large size, petrol, EURO 5 {RER}*<br>*Transport, passenger car, medium size, petrol, EURO 5 {RER}*<br><br>Assumptions:<br><br>• Lifetime: 250,000 km<br>• Large (2000 kg, 8.0–6.0 L/100 km average over 2015–2025)<br>• Medium (1600 kg, 6.0–5.0 L/100 km average over 2015–2025)<br>• EURO 6 gasoline vehicles are considered identical to their EURO 5 counterparts<br><br>Small is not considered | *Transport, passenger car, large size, petrol, EURO 3 {RER}*<br>*Transport, passenger car, medium size, petrol, EURO 3 {RER}*<br>*Transport, passenger car, large size, petrol, EURO 4 {RER}*<br>*Transport, passenger car, medium size, petrol, EURO 4 {RER}*<br>*Transport, passenger car, large size, petrol, EURO 5 {RER}*<br>*Transport, passenger car, medium size, petrol, EURO 5 {RER}*<br>*Transport, passenger car, large size, petrol, EURO 6 {RER}*<br>*Transport, passenger car, medium size, petrol, EURO 6 {RER}* |
| Car (plug-in hybrid) | *Transport, passenger car, electric {RER}*<br>*Transport, passenger car, medium size, petrol, EURO 5 {RER}*<br>Assumptions:<br><br>• Lifetime: 250,000 km<br>• Battery lifetime: 200,000 km<br>• 8 kWh battery [60]<br>• Range of 40 km full electric [60]<br>• Driving cycle "SU1": (12.1 kWh + 2.1 L)/100 km [60]<br>• Curb weight: 1780 kg [60] | *Transport, passenger car, plug-in hybrid {FR}* |
| Car (electric) | *Transport, passenger car, electric {RER}*<br>Assumptions:<br><br>• Lifetime: 250,000 km<br>• Battery lifetime: 200,000 km<br>• 50 kWh battery [61]<br>• Range of 300 km [61]<br>• Charging efficiency 90% (battery-to-wheel consumption: 16.7 kWh/100 km, wall-to-wheel consumption: 18.5 kWh/100 km)<br>• Curb weight: 1800 kg | *Transport, passenger car, electric {FR}* |

Table 3 summarizes the modeled transport modes and specific sources.

### 3.3.3. Life Cycle Impact Assessment (LCIA)

The LCIA part (Q matrix) of the model is based on the recommendations of the European Commission [62]. More specifically, three midpoint indicators are chosen to be presented in this study, as their evolution is most relevant with regard to the sustainability goals identified in Section 1. The climate change midpoint indicator in kilotons of $CO_2$ equivalents (kt $CO_2$ eq) and the respiratory inorganics midpoint indicator in disease incidences, correspond to the goals of (1) combating climate change and (2) improving air quality. In addition the resource use, minerals and metals midpoint indicator in kilograms of Antimony (Sb) equivalents (kg Sb eq) is presented.

## 4. Results

This section reports a summary of the results at different stages of the full model. First, the results of the cohort model predicting the different shares of powertrains within the car fleet of the cross-border commuters for each year of the study period are presented. Next, the mode choice results for commuting related trips and non-work-related trips for both scenarios. Then, the aggregated results of the ABM part of the model showing the travelled distances in (pkm) per transport mode for a single work day over the course of the study period. Both results together allow to derive the final demand vector (2). Finally, the results for three midpoint indicators are presented.

### 4.1. Cohort Model

Using the market shares in Figure 4 as inputs, Figure 6 shows the resulting fleet compositions over time. One can see that EVs show a significant stronger rise in the TIR scenario as compared to the ADEME scenario. While for the latter, the cohort model predicts that there will still be a substantial share of ICEVs in 2050 (Figure 6a), for the former ICEVs seem to almost phase out in 2050 (Figure 6b).

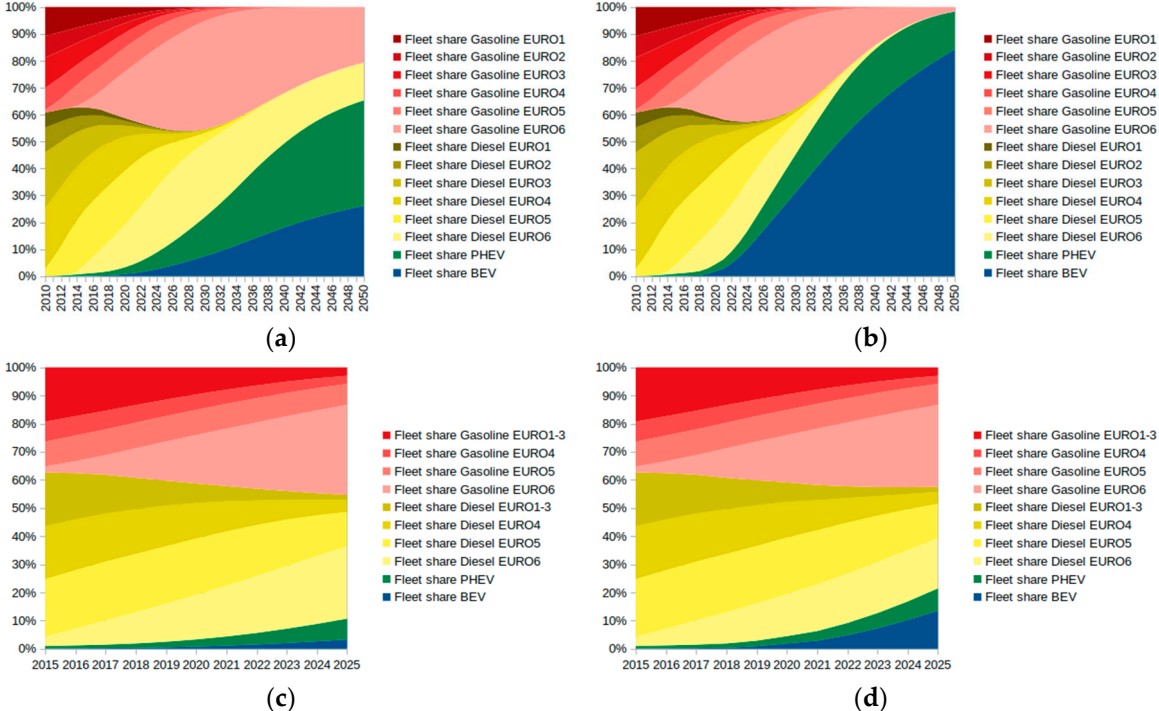

**Figure 6.** Fleet composition shown for both market share scenarios. (**a**) ADEME-inspired fleet shares until 2050; (**b**) TIR-inspired fleet shares until 2050; (**c**) ADEME-inspired fleet shares over the study period; (**d**) TIR-inspired fleet shares over the study period.

However, when only taking the actual study period into account, as depicted in Figure 6c,d, both scenarios only show substantial differences towards the end of the study period, where EVs in the

TIR-inspired scenario make up roughly twice as many vehicles as compared to the ADEME-inspired scenario. One can also see that vehicles following the EURO3 and older emissions standards are phasing out over the study period, resulting in a "cleaner" internal combustion engine fleet by 2025 made up mostly by EURO6 standard vehicles.

### 4.2. Activity-Based Mode Choice Behavior

Figure 7 shows the mode choice behavior over the course of the study period. A distinction is made between mode choice for commuting trips (where activity types for origin and destination are "Home-Work" or "Work-Home") and non-work-related trips, where neither the origin nor the destination activity is of type "Work". This distinction allows testing the specific policy goals formulated for commuter mode choice behavior, e.g., to increase by 50% the number of public transport compared to 2017 as formulated in [7].

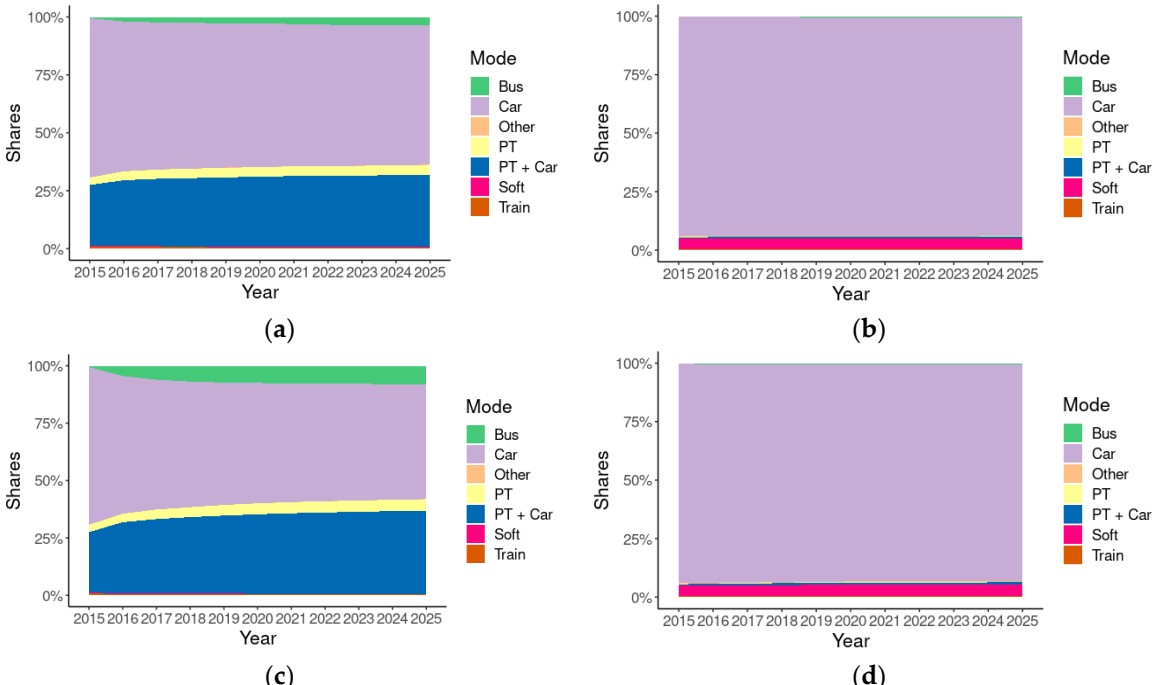

**Figure 7.** Mode choice behavior under policy scenarios for different activity types. (**a**) Commuting modes choice in BAU; (**b**) mode choice for non-work-related trips in BAU; (**c**) commuting modes choice in GREEN; (**d**) mode choice for non-work-related trips in GREEN.

Figure 7a shows the evolution of commuting mode shares for the BAU scenario, revealing a small increase of the share public transport modes (bus, train, multi-modal public transport and public transport + car). This suggests the coverage and speed algorithms applied to the OD matrices have been able to change the mode choice behavior for the targeted trips to some extent. Figure 7b on the other hand shows that non-work-related trips have remained virtually unaffected over the course of the study region.

Figure 7c,d show the same results for the GREEN scenario; however, the mode choice behavior for commuting trips show higher changes with the share of public transport increasing more than for the BAU scenario. Mode shares for non-work-related trips are similar to the BAU scenario.

### 4.3. System Final Demand

The results of the ABM part of the model are shown in Figure 8. Compared to the BAU scenario, the GREEN scenario achieves a reduction of car usage by around 800,000 [pkm], which shifts to bus (around 500,000) and train (around 200,000) by the end of the study period. However, most of this shift

seems to occur during the first years of the study period, while for later years increasing demand of the growing population is mostly satisfied by car usage.

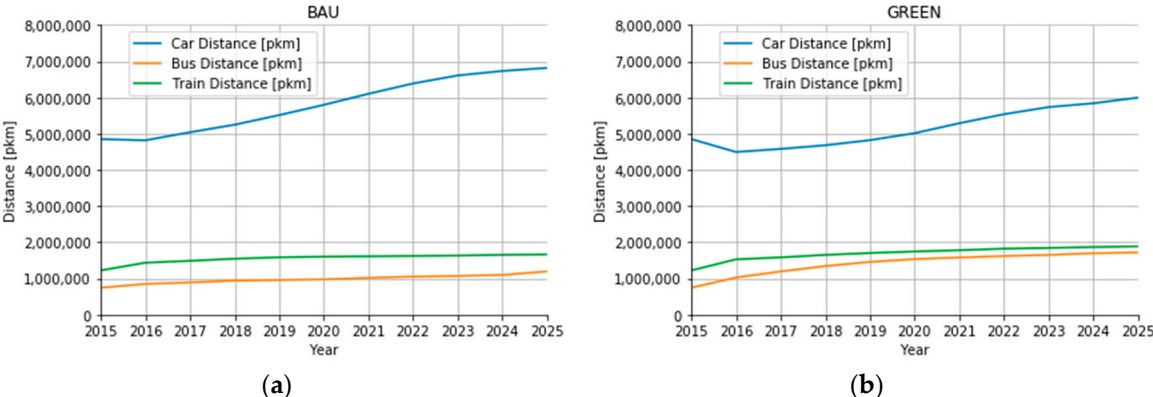

**Figure 8.** Cumulative travel distances over time per mode. (**a**) BAU; (**b**) GREEN.

### 4.4. Climate Change

The final results of the ABM/LCA coupled model with regard to climate change are presented in Figure 9. Figure 9a,b show the results where the parameters listed in Table 1 remain at the level of 2015, while Figure 9c,d show the results where the parameters listed in Table 1 evolve linearly between 2015 and 2025.

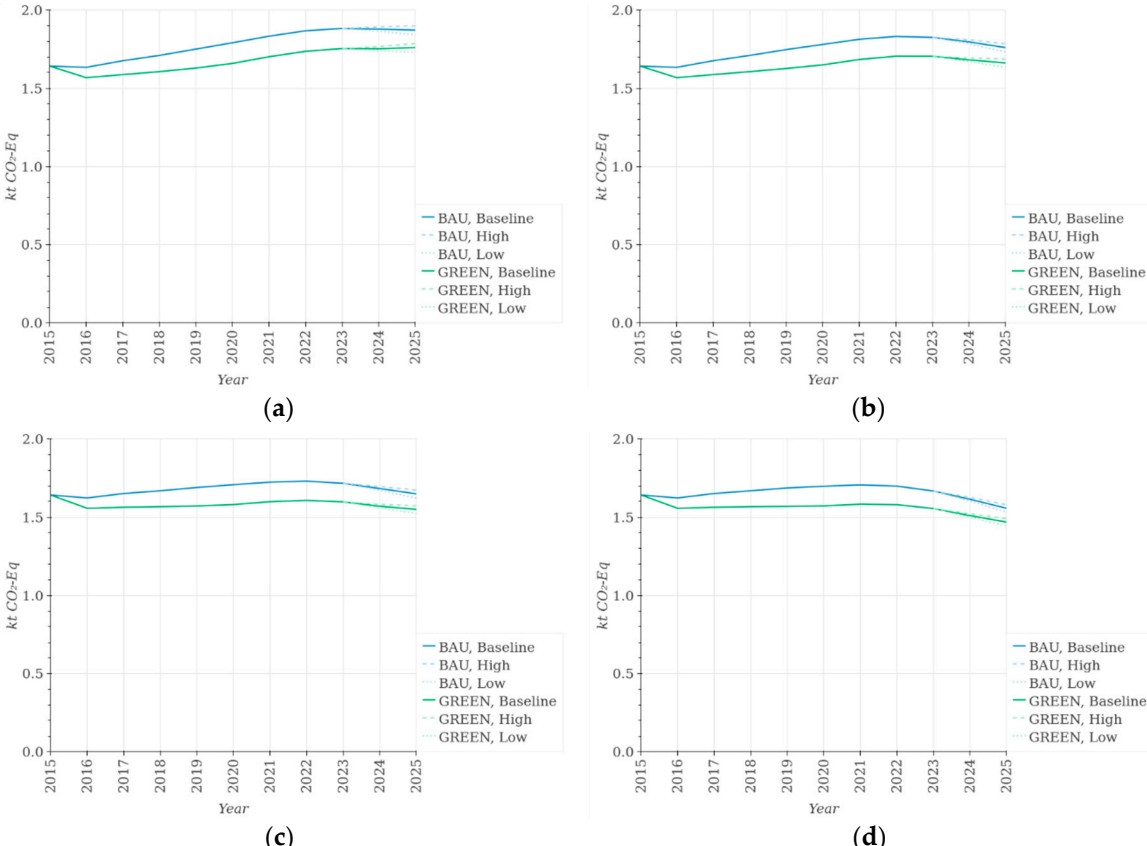

**Figure 9.** Climate change related impact for a single work day over the course of the study period. (**a**) ADEME-inspired EV scenario and fixed parameters; (**b**) TIR-inspired EV scenario and fixed parameters; (**c**) ADEME-inspired EV scenario and evolving parameters; (**d**) TIR-inspired EV scenario and evolving parameters.

There is evidence that for each of the figures the GREEN scenarios perform better than the BAU scenarios. This can easily be explained by the lower related impacts of public transport modes compared to the individual car. Additionally, until at least 2021, all scenarios predict an increase of climate change impacts compared to 2015 for a single work day. This can be explained by the strong increase of the total number of French cross-border commuters commuting to Luxembourg over the study period, as well as the slow initial increase of EVs in the car fleet regardless of the scenario.

Figure 9d shows a reversal of the trend starting 2021, followed by Figure 9b,c in 2022 both for the GREEN and BAU scenarios. Only Figure 9a does not show a clear reversal of the trend of increasing impacts, although they seem to stabilize in 2023. We can conclude that a combination of measures on the transport system, as well as decarbonizing the electricity mix can potentially allow to counteract the additional impact due to external drivers such as economic growth in the long term. Electric mobility alone, without additional measures such as the decarbonization of the electricity mix or a shift towards public transport modes does not allow to mitigate the impact of the increasing population until 2025.

Finally, for the GREEN scenario the impacts in 2016 are always lower than in 2015. This can be explained in part by some data scarcity, where some important existing public transport links might not be present in that data, as well as the prioritization of communes with a large number of cross-border commuters when manipulating the OD matrices.

### 4.5. Respiratory Effects

Similar to the last section the environmental impact related to respiratory effects are presented by distinguishing four experimental frames.

In Figure 10a, the ADEME-inspired scenario for electro mobility is chosen while the LCI parameters remain on the 2015 levels. Regardless of the population scenario for both BAU and GREEN, a steady increase of impacts is observed until 2023, after which a slight decrease is observed.

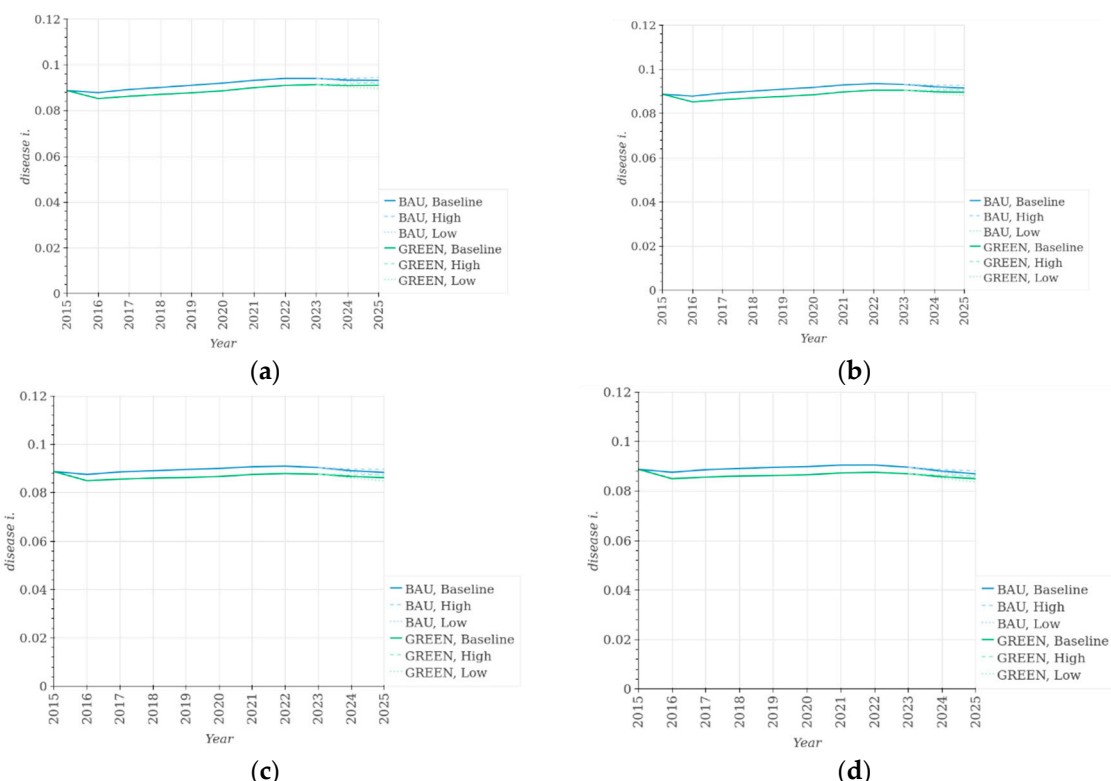

**Figure 10.** Respiratory effects related impact for a single work day over the course of the study period. (**a**) ADEME-inspired EV scenario and fixed parameters; (**b**) TIR-inspired EV scenario and fixed parameters; (**c**) ADEME-inspired EV scenario and evolving parameters; (**d**) TIR-inspired EV scenario and evolving parameters.

In Figure 10b, the more ambitious scenario for electric mobility is adopted, while the LCI parameters remain unchanged. For both BAU and GREEN the impacts increase until 2022, only after the increasing share of EVs in the car fleet having lower related impacts allow to reverse the trend for all population scenarios.

In Figure 10c, the influence of the evolving LCI parameters can be observed for the ADEME-inspired scenario for electric mobility. Again, for both BAU and GREEN, a steady increase for the most part of the study period is found; towards the end, cleaner electricity mixes allow to reverse the trend and, at least for the GREEN scenarios and BAU low population scenario, reduce impacts compared to 2015.

Finally, in Figure 10d the TIR-inspired scenario for electric mobility is adopted while LCI parameters evolve over the course of the study period. Unsurprisingly, the combination of a cleaner electricity mix and a high share of EVs allows for the best results showing a slightly stronger trend than Figure 10c of impact mitigation after 2022.

### 4.6. Minerals and Metals Depletion

In Figure 11a–d the mineral and metal depletion related impacts are depicted. This third impact category paints a different picture compared to the two former ones. While before we have highlighted the positive impacts of technological policy measures, this third category shows that in some cases, a higher share of EVs in the case fleet can lead to an increase of environmental impacts. As the battery of EVs requires several scarce materials (e.g., lithium and rare earths), a higher share of EVs causes higher metal depletion related impacts. In consequence Figure 11b,d show consistently higher impacts compared to Figure 11a,c, respectively.

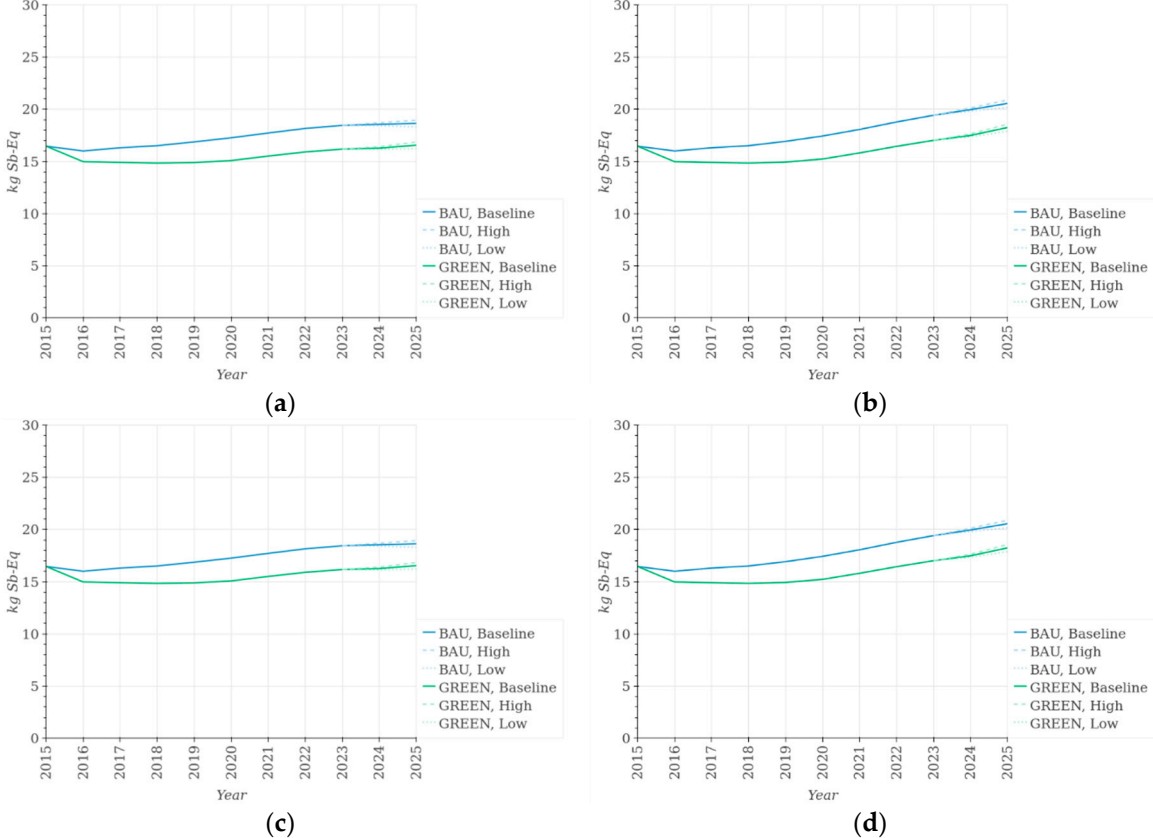

**Figure 11.** Mineral and metal depletion related impact for a single work day over the course of the study period. (**a**) ADEME-inspired EV scenario and fixed parameters; (**b**) TIR-inspired EV scenario and fixed parameters; (**c**) ADEME-inspired EV scenario and evolving parameters; (**d**) TIR-inspired EV scenario and evolving parameters.

In order to show a full picture of all impacts, we provide a comprehensive overview of several additional impact categories in the supplementary materials.

## 5. Discussion

In this section the findings of our study are discussed under three broad contexts, by dividing the section in three distinct parts. In the 'Promises' section, we aim at discussing to what extent the goal of the study was reached and the added value of the scientific knowledge created using such an approach. In the 'Challenges' section, we aim at discussing the current limitations of the approach as well as data scarcity which hinders us from making concrete policy recommendations at the present time. Finally, in the 'Prospects' section, we aim at discussing how the distinct challenges could be overcome and how some of the identified limitations of the present work are planned to be addressed in future research endeavors.

Before looking at these three aspects of our work, we aim at putting the presented results in the context of results of prior studies found in the literature review by qualitative comparison. Most studies confirm that results regarding the fleet shares of EVs. Cellura et al. [16] showed that scenarios with higher shares of EVs reduce both the global warming potential and particulate matter emissions when combined with renewable energy sources. On the other hand, Cellura et al. [16] confirmed an increase of resource depletion (mineral, fossil and renewable resource depletion in their case). Several authors [20,23,27,29,32] showed similar results for climate change mitigation.

Garcia et al. [31] showed that achieving high shares of EVs in a car fleet can take time and conclude that only after 2025 they will become significant. This is in accordance with the results of our cohort model. With regard to global warming potential, their results showed that the potential reduction depends on several parameters such as their market share and the electricity mix.

Our results contrasting the BAU and GREEN scenario confirm findings from, e.g., [33] and [15], in that a shift towards public transportation can reduce global warming potential as well as other impact categories.

### 5.1. Promises

The first goal of coupling ABM and LCA, as mentioned in Section 1 of the present work, was to illustrate how the framework allows to assess the success of policy action in reaching specific targets on the trip level (i.e., mode share for commuting related trips). In consequence, we have defined policy scenarios for public transport accordingly and our results regarding the activity patterns in Section 4.2 illustrate how the modeling framework allowed to reach this goal. We believe that such data can be of value for policy makers when faced with decisions regarding concrete alternative policy measures to improve public transport quality. In addition, our framework allows for spatially explicit data with regard to mode choice as illustrated in Figure A1 (Appendix A) showing the share of public transport in 2015 and in 2025 for both policy scenarios. One can see that while the BAU scenario is able to increase the share of public transport usage for some communes (e.g., the large cities Thionville and Metz for which mainly the speed algorithm is relevant), the GREEN scenario reaches a wider area (result which can be attributed to the coverage algorithm for which no threshold was applied).

The second goal of coupling ABM and LCA was to illustrate how life cycle impacts of the entire transport system can be calculated for different policy scenarios. By combining both public transport scenarios with market scenarios regarding public transport and external drivers, it allowed to assess the system's final demand for life cycle processes and derive the final demand vectors over the study period. This led to characterize the trend of environmental impacts for a single work day in Sections 4.4 and 4.5 for three midpoint impact categories which are relevant to policy makers with regard to international agreements (Paris climate agreement and air quality commitments) and resource depletion.

Results suggest that combinations of measures can be effective to reduce climate change-related impacts at least in the long run if decisions are made now. Results in Sections 4.4 and 4.5 point out that stronger investments in public transport only have an effect in the early years of the study period,

while for later years the curves are no longer diverging. This is mainly due to our scenario definition, where both types of algorithm (coverage and speed) are first applied to the communes with a high share of cross-border commuters and thus measures in later years seem to have a negligible effect. On the other hand, if high market shares of EVs could be reached in the mid and long term, especially if accompanied with a de-carbonization of the electricity mix, the high rise in cross-border commuters can be mitigated.

*5.2. Challenges*

However, the integration of ABM and LCA comes with some specific challenges, which hinder us from making concrete policy recommendations. As both approaches are data intensive, data quality and scarcity can be encountered at different stages of the model development. High quality and recent survey data for the specific study region, which is required to train the ABM and to generate a synthetic population, can be hard to come by. Legitimate data protection concerns, cost or low response rates are just some of the reasons. In addition, high quality data describing regional cross-border transport systems is often not available. While for some parts of a study region open data initiatives might exist, for other parts such data must be requested either from the authorities or in some cases individual operators. Different existing formats (e.g., GTFS [63] and neptune [64]) might hinder easy data merging. Similar issues can be encountered when building the LCI, where public transport bus and train fleet data and manufacturer data needs to be collected; however, this issue is not particular to the ABM/LCA coupling.

With regard to the modeling itself, several limitations of our current implementation have to be acknowledged. Some of them emerge from data scarcity and quality issues described above. Currently public transport policy measures are only implemented with regard to travel time and coverage of the public transport system, while, e.g., the impact of reducing travel cost is not tested. Several simplifications are made when deriving the system's final demand, most noticeably making assumptions with regard to public transport occupancy rates, which can vary strongly over time (peak versus off-peak) or between different lines. We also did not account for constraints such as public transport capacity and the impact of congestions on travel time. Finally, we did not account for the impact of public transport evolution on long term choices such as car ownership, transit pass ownership, work and resident location. We also did not account for the impact of public transport evolution on other aspects of the activity pattern, such as location choice. These limitations, in our view, constrained the mitigation impact of measures aiming at behavioral change in our study. We therefore do not compare the impact of behavioral change against technological change.

The limitations of the life cycle assessment model arise both from the inventory modeling and impact assessment. Regarding life cycle inventory construction, challenges typically arise when compiling specific data for a more generic case [65]. For example, the choice of splitting ICE vehicles by powertrain and segment is based on data availability; whereas electric vehicles are represented by a single inventory deemed to be representative of the average fleet. Impact assessment adds another layer of uncertainty, as impact assessment models are generic, both in time and space, and—in our case—do not represent exactly the environmental effects in Luxembourg from 2015 to 2025. It should also be noted that both life cycle inventories and impact assessment methods do not cover all substances in an exhaustive manner.

Finally, uncertainty analysis and model validation need to be addressed. Reference [47] reviewed the potential methodologies towards uncertainty analysis of such models extensively and provided a first classification of uncertainty sources. The main challenge for the present framework is a joint propagation of uncertainty forthcoming from inputs (e.g., estimated travel time data) and parameters (both for the ABM and LCA part), as well as the uncertainty from individual choices (e.g., mode choice).

Model validation has been addressed for ABM in the literature, where methods range from validation of the models explanatory power of aggregated measures (e.g., mode shares), over methods investigating sub-model performance separately [45], to more sophisticated methods deriving

multidimensional sequence alignment based distance measures between the predicted and observed activity patterns [66]. Mariante [45] provided validation for each of the sub-models of the ABM used in our study. For LCA on the other hand, validation in the sense of empirical testing of the overall outcome is often impractical [67]. Uncertainty analysis, sensitivity analysis and data quality thus gain importance.

### 5.3. Prospects

To fully take advantage of the potential of the presented framework, some of the challenges in the last section must be addressed in order to make sound predictions for policy support. To start, the current trend of open data (as illustrated by the Luxembourgish open data initiatives [53]) need to be continued to assure that high quality and recent data is available to model commuter behavior and transport systems. New avenues, such as co-operations between local or regional authorities and research to collect and use live user data should be pursued [68].

To improve our current model, such data could be used to assess the impact of more recent policy measures such as making the public transport system free in Luxembourg, or evaluate individual infrastructure measures such as the new tramway line in the capital city and plans to expand it to the second largest city in the south of Luxembourg (Esch-sur-Alzette) [7].

Model improvement could also be guided by information about the contribution to the output uncertainty of individual parameters or sub-models. This could allow to steer future data collection efforts and model development. Currently, work is ongoing to estimate the uncertainty forthcoming from each of the ABM sub-models and its impact on the overall model outcomes. Different methods of model validation (validation of aggregated outputs versus multi-dimensional sequence alignment) are planned to be applied to the ABM part of the model, while for the LCA part data quality measures and uncertainty analysis are planned to be applied.

Finally, the granularity of the model shall be reduced, to arrive at a finer one, to be able to account for temporal variation of travel times of the course of a work day rather than using static OD matrices. Additionally, prior efforts to evaluate the impact of policies on EV purchases using agent-based modeling, presented in [32], shall be better integrated instead than using market scenarios and cohort modeling.

**Supplementary Materials:** The supplementary materials to this article includes (1) a complete list of all articles and exclusion criteria of the PRISMA review, (2) the code of the mode choice model of the ABM and (3) the results for a comprehensive set of midpoint indicators.

**Author Contributions:** Conceptualization, P.B., T.N.G., T.G., P.G. and E.B.; Data curation, P.B., T.N.G. and S.K.; Funding acquisition, P.G. and E.B.; Investigation, P.B., T.N.G. and T.G.; Methodology, P.B., T.-Y.M., G.L.M., S.K. and E.B.; Project administration, E.B.; Software, P.B., T.N.G., L.C., T.-Y.M., G.L.M. and P.G.; Supervision, E.B.; Validation, P.B., T.N.G., T.G., L.C., T.-Y.M. and G.L.M.; Visualization, P.B.; Writing—original draft, P.B., T.G. and L.C.; Writing—review & editing, P.B., T.N.G., T.G., T.-Y.M. and E.B.

**Funding:** This research was funded by Fonds National de la Recherche Luxembourg, grant C14/SR/8330766.

**Acknowledgments:** We acknowledge Frédéric Schmitz for his work providing the survey data used to create the synthetic population, model parameter estimation, as well as the formal analysis of the data. The survey database relies on the work of the CABaC 2010–2013 research project (Construction and Analysis of a Knowledge Base on mobility habits and attitudes towards energy of cross-border workers in Luxembourg, FNR INTER/CNRS/09/01). We acknowledge Javier Babí Almenar for his contribution to creating the spatial figures.

**Conflicts of Interest:** The authors declare no conflict of interest.

## Appendix A

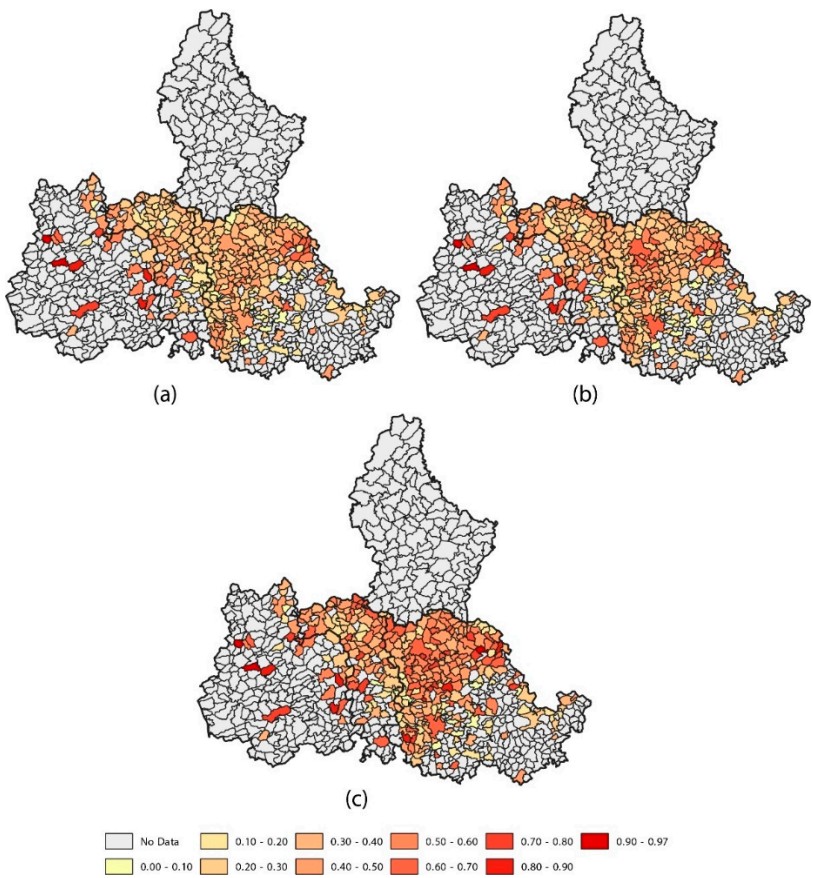

**Figure A1.** Spatial distribution of public transport mode shares for commuting. (**a**) Reference year; (**b**) BAU in 2025; (**c**) GREEN in 2025.

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
