# Peer review of "Coupling Activity-Based Modeling and Life Cycle Assessment—A Proof-of-Concept Study on Cross-Border Commuting in Luxembourg"

_sustainability, doi:10.3390/su11154067_

Round 1
Reviewer 1 Report
The paper "Coupling activity-based modelling and Life cycle assessment – A proof-of-concept study on cross-border commuting in Luxembourg " analyzes an interesting topic, linking LCA and activity-based modelling.
The paper is bases on LCA, but does not show information regarding LCA following common scientific practice as stated by ISO 14040 and 14044. Both ISO standards should be followed and cited in the paper.
Several comments to further improve the paper:
- Authors must follow the instructions for authors and the template.
- http://www.who.int/mediacentre/news/releases/2014/air-pollution/ and all the webpages should be cited as references.
- A proper review of the state of the art regarding life cycle impacts of mobility policies is expected
-Figure 1, complete the 24hour cycle establising the duration of Home
-If this study is based on LCA, a proper definition of the LCA study must be included. This is currently lacking in the paper. Functional unit, system boundaries, environmental impact categories, LCI database... as ISO 14040 and 14044 require, and explained in material and methods subsection
- Table 3, why euro5? euro 6 registration is 2015
- Table 3, check electic bus average consumption, as 1.3kWh/100 km is not wrong.
- Table 3, petrol car tend to be lighter than diesel
- Table 3, electric cars tend to be heavier than ICE cars due to battery weight
- An ICE car similar to a Nissan leaf does not weght 2000 Kg
- Why do figures show PHEV, when these are not included in Table 3?
- The selection of only two midpoint categories is a clear limitation of this study and may bias the results, as those categories do not reflect the environmental impact created by the batteries.
- The limitations of the study must be clearly explained.
Author Response
The paper is bases on LCA, but does not show information regarding LCA following common scientific practice as stated by ISO 14040 and 14044. Both ISO standards should be followed and cited in the paper.
While the paper is only partly based on LCA, we understand the reviewer's comment about the description of the life cycle model. We added several paragraphs in order to be in line with the ISO standard.
Authors must follow the instructions for authors and the template.
To the best of our knowledge we followed the instructions and template (with the exception of citing webpages as footnotes). If there are any other incoherences, please make them explicit.
http://www.who.int/mediacentre/news/releases/2014/air-pollution/ and all the webpages should be cited as references.
We made sure that all webpages are now cited as references instead of footnotes.
A proper review of the state of the art regarding life cycle impacts of mobility policies is expected
We now conducted a systematic review of the state of the art. We document the review using the PRISMA flowchart and provide all identified records (266) in the supporting information. We reviewed the 28 retained articles distinguishing four categories of policy measure scenarios and their combinations.
Figure 1, complete the 24hour cycle establising the duration of Home
We completed the 24h cycle as suggested.
If this study is based on LCA, a proper definition of the LCA study must be included. This is currently lacking in the paper. Functional unit, system boundaries, environmental impact categories, LCI database... as ISO 14040 and 14044 require, and explained in material and methods subsection
We have restructured the LCA section to align with the ISO standard definition, with subsections "Goal and scope", "LCI" and "LCIA". Interpretation of the results is the object of the "Results" section.
Table 3, why euro5? euro 6 registration is 2015
This is correct. We have created EURO6 processes. We also now distiguish between all emission standards in the cohort model and distinguish EURO3, EURO4, EURO5 and EURO6 in the LCA.
Table 3, check electic bus average consumption, as 1.3kWh/100 km is not wrong.
Thank you for pointing the error, consumption is 1,3 kWh/km. Corrected.
Table 3, petrol car tend to be lighter than diesel
This is true, although the difference is marginal. Bauer et al. (2015) report average weights of 1542 and 1560 kg for gasoline and diesel cars, respectively. For medium vehicles, we have changed our values back to the original ecoinvent assumption of 1600 kg.
Table 3, electric cars tend to be heavier than ICE cars due to battery weight
We now use an average value of 1800 kg for electric vehicles, with a 50 kWh battery and a 300 km range. This is to account for the diversity of the electric car market - a Renault Zoé (41 kWh) weighs 1468 kg while a Tesla S60 weighs 1900 kg (and more in 75, 90 kWh version...).
An ICE car similar to a Nissan leaf does not weght 2000 Kg
See comments above. The electric vehicle inventory is representative of the whole fleet, while the ICE vehicles are modelled by segment.
Why do figures show PHEV, when these are not included in Table 3?
As was written in section 3.3: "At the current stage we are not distinguishing BEVs and PHEVs [...]" for the LCI phase. We now included a PHEV inventory and conduct the LCA accordingly.
The selection of only two midpoint categories is a clear limitation of this study and may bias the results, as those categories do not reflect the environmental impact created by the batteries.
We agree that this paints a biased picture. The initial choice of midpoints refected to interests of policy makers based on the review of strategic documents (see introduction). However, we agree that a more comprehensive set of midpoint indicators should be presented. To this end we now added the results of all midpoints (which are not in the manuscript) to the supporting informations. We also added the midpoint "resource use, mineral and metal" which reflects impacts related to battery production to the results section.
The limitations of the study must be clearly explained.
We added paragraphs explaining limitations of the study (with regard to the ABM and LCA part) to the discussion section.
Reviewer 2 Report
Coupling activity-based modelling and Life cycle assessment – A proof-of-concept study on cross-border commuting in Luxembourg
The study does exactly what the title says, i.e. uses a multi-model approach to assess the impact of different policies to reduce the impact of commuting in Luxembourg. This is a strong paper and I have only some minor comments. The modelling is very complex and hardly reproducible, so my main recommendation to the Authors is to improve the clarity of the methods section, and to extend the discussion section. In this way the reader should get a better idea of how the models interact and based on what data, and how to interpret the results in the light of model complexity and uncertainty. Specific comments on following.
Line 96. My guess is that the target audience of this study would not be too familiar with ABM so I recommend spending some words here to briefly describe what ABM is about and how it works in general.
Line 107. Perhaps this is a matter of personal taste, but I find redundant the use of acronyms like PT and CBC. The Authors could consider simply using the full wording (Public transport and cross-border commuters) to increase readability. OK to keep standard acronyms like LCA and ABM
Lines 108-110. What is a bit unclear here is why exactly these two options are considered, given that current policies include several different measures as mentioned in the introduction. Section 3.1. has the same problem. Please clarify.
Equation 1. P means "pattern"? Please define. Why is this defined as a ration to the reference category and what is "other" exactly? I think the Authors could better explain how the dependent variable on the left-hand side of the equation should be interpreted and what its meaning is. Another way to clarify this even further would be to provide an example of the input data and result (with units).
Line 150. "one simulation of the ABM as rM." Meaning of this? What does r and M stand for? Please clarify.
Lines 187-194. I appreciate that the authors provide the R code for the ABM in SI. However, without the rest of the python code and a sample of the input data this is not very useful. Brightway is open source software so it's not clear what is under IPR protection and why the code can't be provided. Please consider enriching the SI with more code and data, or at least link to a repository.
Line 302. Specify it's cutoff version.
Line 303. Does this mean that the model assumes an increase in domestic production capacity and a decrease in imports? Using "displace" can be confusing. Please clarify.
Figure 6. These trends seem surprisingly flat compared to Figure 5. Does it mean that behavioral change is much more difficult to achieve than tech change - in this specific case at least? This seems a very interesting point for discussion and I invite the authors to elaborate on that in Section 5.
Lines 380-383. This should be in the methods section, not in the results section.
Line 398. Good conclusion
Section 5. In this section I miss these things: 1) the comparison with results of previous studies. 2) a discussion of the modelling uncertainties. 3) A discussion of the specific and general validity of these results (how can results be validated for this case and to what extent can be extended to other contexts?)
Lines 493-500. This seems a marginal issue. The authors have not provided the full model code, so I assume they are the only ones capable to reproduce these results. How high is the chance that this model should be run on-the-fly at a workshop, really? I would rather reflect on how this work can be replicated and validated, and what is the reproducibility of these results.
Author Response
The study does exactly what the title says, i.e. uses a multi-model approach to assess the impact of different policies to reduce the impact of commuting in Luxembourg. This is a strong paper and I have only some minor comments. The modelling is very complex and hardly reproducible, so my main recommendation to the Authors is to improve the clarity of the methods section, and to extend the discussion section. In this way the reader should get a better idea of how the models interact and based on what data, and how to interpret the results in the light of model complexity and uncertainty. Specific comments on following.
We have addressed the specific comments below relating to this broader comments.
Line 96. My guess is that the target audience of this study would not be too familiar with ABM so I recommend spending some words here to briefly describe what ABM is about and how it works in general.
We added a two sentences introducing ABM and referring to relevant literature.
Line 107. Perhaps this is a matter of personal taste, but I find redundant the use of acronyms like PT and CBC. The Authors could consider simply using the full wording (Public transport and cross-border commuters) to increase readability. OK to keep standard acronyms like LCA and ABM
We addressed the comment as recommended for the acronyms PT and CBC.
Lines 108-110. What is a bit unclear here is why exactly these two options are considered, given that current policies include several different measures as mentioned in the introduction. Section 3.1. has the same problem. Please clarify.
To make the rational behind this choice more explicit we added a paragraph to the introduction section and a sentence to section 3.1. The main reasons are: (1) both measures are explicitly mentioned in most of the reviewed strategic policy documents; (2) they reflect both technical and behavioral solutions.
Equation 1. P means "pattern"? Please define. Why is this defined as a ration to the reference category and what is "other" exactly? I think the Authors could better explain how the dependent variable on the left-hand side of the equation should be interpreted and what its meaning is. Another way to clarify this even further would be to provide an example of the input data and result (with units).
P stands for "probability", where P(M_ij=m) is the choice probability of mode m for individual i and activity j. We added two sentences to the paragraph to clarify.
Line 150. "one simulation of the ABM as rM." Meaning of this? What does r and M stand for? Please clarify.
M is a concrete instance of the ABM part of the simulator relating to an experimental frame (including year and scenario). We added a sentence to clarify the meaning.
Lines 187-194. I appreciate that the authors provide the R code for the ABM in SI. However, without the rest of the python code and a sample of the input data this is not very useful. Brightway is open source software so it's not clear what is under IPR protection and why the code can't be provided. Please consider enriching the SI with more code and data, or at least link to a repository.
The R code we have provided initially is useful from our point of view, since it provides the exact manner to estimate and apply the choice model, for a given population. Our commitment is that any other researcher or interested party that wishes to collaborate to further improve or investigate our findings is welcome, and under a mutually agreed legal basis, we are available to work in collaboration.
Line 302. Specify it's cutoff version.
We specified the database version. We also added that it corresponds to a cradle-to-grave approach.
Line 303. Does this mean that the model assumes an increase in domestic production capacity and a decrease in imports? Using "displace" can be confusing. Please clarify.
The word "displace" has been removed, and the sentence reformulated.
Figure 6. These trends seem surprisingly flat compared to Figure 5. Does it mean that behavioral change is much more difficult to achieve than tech change - in this specific case at least? This seems a very interesting point for discussion and I invite the authors to elaborate on that in Section 5.
Indeed, achieving behavioral change can be difficult. However, over the course of the study period (2015-2025) the differences don't seem that striking in figure 5 (c) and (d) and firgure 6. In addition, mode choice is also influenced by more long term choices which are not explicitly modelling in our framework yet, e.g., car ownership remains constant for the synthetic population. Taking into account the impact of a better PT system on car ownership in the long term could reveal higher potential for mode shift and climate change mitigation. We now raise this among other limitations in the discussion.
Lines 380-383. This should be in the methods section, not in the results section.
We moved this part to the methods section. Note that it also relates to comment 2.06.
Line 398. Good conclusion
Section 5. In this section I miss these things: 1) the comparison with results of previous studies. 2) a discussion of the modelling uncertainties. 3) A discussion of the specific and general validity of these results (how can results be validated for this case and to what extent can be extended to other contexts?)
1) To the best of our knowledge there are no previous studies that have examined the study region. However we qualitatively compare the results to prior studies found in the literature review. 2) We added a paragraph discussing uncertainties (please note that a separate manuscript is being prepared to address uncertainty analysis of ABM/LCA coupled models). 3) We added a section discussing the validation of the individual model parts.
Lines 493-500. This seems a marginal issue. The authors have not provided the full model code, so I assume they are the only ones capable to reproduce these results. How high is the chance that this model should be run on-the-fly at a workshop, really? I would rather reflect on how this work can be replicated and validated, and what is the reproducibility of these results.
We removed the paragraph and focussed on validation and uncertainty as suggested above.
Round 2
Reviewer 1 Report
The authors have properly replied to all my comments. My recommendation is accept.